# Effect of Composition and Viscosity of Spinning Solution on Ultrafiltration Properties of Polyphenylene Sulfone Hollow-Fiber Membranes

**DOI:** 10.3390/membranes12111113

**Published:** 2022-11-08

**Authors:** Tatyana Anokhina, Alisa Raeva, Stepan Sokolov, Alexandra Storchun, Marina Filatova, Azamat Zhansitov, Zhanna Kurdanova, Kamila Shakhmurzova, Svetlana Khashirova, Ilya Borisov

**Affiliations:** 1A.V. Topchiev Institute of Petrochemical Synthesis RAS, 29 Leninsky Prospekt, 119991 Moscow, Russia; 2Progressive Materials and Additive Technologies Center, Kabardino-Balkarian State University Named after H.M. Berbekov, St. Chernyshevsky, 173, 360004 Nalchik, Russia

**Keywords:** polyphenylene sulfones, molecular mass, polymeric hollow fiber membranes, viscosity, wet spinning method

## Abstract

In this work, PPSUs with different molecular weights were synthesized for the development of highly permeable ultrafiltration hollow fiber membranes for the first time. The *M*_W_ of the synthesized polymers was controlled by varying the monomers molar ratio within 1:1–1.15 under the same synthesis conditions. Based on the study of the rheological properties of polymer solutions, a high molecular weight PPSU (*M*_W_ = 102,000 g/mol) was chosen for the formation of hollow fiber membranes. The addition of PEG400 to the spinning solution led to an increase in viscosity, which makes it possible to work in the region of lower PPSU concentrations (18–20 wt. %) and to form membranes with a less dense porous structure. With the addition of PEG400 to the spinning solution, the membrane permeance increased sharply by more than two orders of magnitude (from 0.2 to 96 L/m^2^·h bar). At the same time, the membranes had high rejection coefficients (99.9%) of Blue Dextran model filtered substance (*M*_W_ = 69,000 g/mol).

## 1. Introduction

Estimates of the growth of the Earth’s population indicate that by 2023 the number of people on our planet will exceed eight billion. At the same time, one of the global problems will be the lack of clean drinking water for most people [1,2]. Human activities are the main cause of water quality decline. Poor-quality wastewater treatment, failure of septic systems, and livestock waste are among the main sources of water pollution with biogenic waste [2]. The World Health Organization (WHO) has stated that around 785 million people are concerned about the availability of clean drinking water today, and two billion people are already using contaminated water sources for drinking [3]. In order to obtain purified drinking water, along with heavy metals, biological contaminants, including bacteria and viruses, must be removed from it. In this regard, membrane filtration is becoming one of the most popular water treatment technologies. Membrane processes have a number of advantages over traditional approaches. These are modularity, compactness and stability in time [4]. Microfiltration (MF) and ultrafiltration (UF) membranes [5] are the most effective for removing water-polluting components, including bacteria and viruses [2,6,7].

Ceramic membranes based on alumina, silicon dioxide, titanium dioxide, zirconium dioxide and silicon carbide have high chemical and thermal stability and are effective for filtration and subsequent sterilization. The disadvantages of ceramic membranes are high cost and fragility [8]. Polymeric materials, such as polyvinylidene difluoride, polysulfone, polyamide, polyethersulfone, and cellulose acetates, currently predominate the membrane market due to their low cost, as well as ease of preparation and scalability [5,9,10]. Additional requirements, primarily the ability to regenerate [10] and sterilize [8], are imposed on membranes when filtering media containing pathogenic organisms. For example, the most common and cheapest method of sterilizing filtration membranes is treatment with superheated steam at a temperature of more than 100 °C. Not many commercial polymeric membranes can withstand such processing repeated regularly for long-term work. For example, highly chemically resistant polymers, such as polysulfone and polyethersulfone, can withstand no more than 100 steam sterilization cycles, according to the polymer manufacturer BASF.

In this regard, polyphenylene sulfone (PPSU) is a promising polymer material for the production of filtration membranes with unique properties: high long-term thermal stability, high mechanical strength, increased resistance to hydrolysis, plasticization and cracking under the action of a number of organic solvents compared to other polymers [11,12,13]. The glass transition temperature of PPSU (220 °C) is higher than that of polysulfone (PSF) (190 °C) and is close to the glass transition temperature of polyethersulfone (PES) (225 °C). According to the manufacturer of PPSU (BASF), this polymer does not change its characteristics even after 2000 cycles of sterilization with superheated steam. For this reason, it is replacing other polymers in medicine and the pharmaceutical industry. Thus, the work life of PPSU membranes is practically unlimited. The development of highly permeable filtration membranes based on polyphenylene sulfone will qualitatively change the understanding of the properties of polymer membranes, both from a fundamental and practical point of view.

To date, a number of works have been described in the literature on the preparation of ultrafiltration membranes based on PPSU [11,14,15,16,17,18,19,20,21,22,23] (Table 1). According to the data given in Table 1, it can be concluded that studies of UF membranes are carried out mainly with commercial PPSUs with a molecular weight of 48,000–65,000 g/mol from manufacturers such as BASF and Solvey. The authors [24] proposed a new method of ultrafiltration membrane formation from PPSU. The authors used systems characterized by the upper critical solution temperature (UCST), the gelation point (TG), and the lower critical solution temperature (LCST). The developed method uses PPSU solutions in the temperature range from LCST to GP. Meanwhile, the temperature of the coagulation bath should be between GP and LCST. Using the proposed method, flat membranes were formed with a water permeance of 486 L/m^2^·h bar and a rejection of human serum albumin (HSA) of 90%. This is the best result for ultrafiltration membranes made of pure PPSU, which are known today [15]. In the latest works of 2020, to spin high-performance hollow fiber (HF) membranes with a permeance of 500–600 L/m^2^·h bar, mixtures of PPSU with multiblock copolymers made from polyethylene oxide and polypropylene oxide segments were used [21,22]. At the same time, one work is described in the literature on the preparation of HF membranes from pure PPSU (Solvay, Alpharetta, GA, USA), and their permeance was 55.8 L/m^2^·h bar [11].

A small number of works on the preparation of HF membranes from pure PPSU may be due to the fact that the commercial PPSUs presented on the market have a molecular weight that is not suitable for the production of HP membranes.

In this regard, the aim of this work was to study the effect of the PPSU molecular weight on the morphology and filtration properties of hollow fiber membranes formed on the basis of this unique polymer.

## 2. Materials and Methods

### 2.1. Materials

For the synthesis of PPSU, reagents manufactured by Sigma Aldrich were used: dihydroxydiphenyl, 4,4-dichlorodiphenyl sulfone, potassium carbonate and dimethylacetamide (DMAc).

To form membranes, N-methylpyrrolidone (NMP) Across Organics 99% extra pure was used as a solvent. A pore-forming agent, polyethylene glycol (PEG400) with *M*_W_ = 400 g/mol and a dynamic viscosity of 120 mPa∙s from Acros Organics, was introduced into the solution.

The transport characteristics of the hollow fiber membranes were studied using distilled water. Model dye Blue Dextran with a molecular weight of 69,000 g/mol (Sigma Aldrich Co. LLC, St. Louis, MO, USA) was used to evaluate the separating properties of hollow fiber membranes.

### 2.2. Synthesis of PPSUs

The synthesis was carried out in a four-necked reaction vessel with a mechanical stirrer, a thermocouple, an inert gas capillary, a Dean-Stark trap, and a reflux condenser. 55.8 g (0.3 mol) dihydroxydiphenyl, 86.1 g (0.3 mol) 4,4-dichlorodiphenyl sulfone, 69.1 g (0.45 mol) potassium carbonate, 470 mL DMAc were charged. The reaction mass was heated to 165 °C with continuous stirring in an inert gas flow. After distillation of water and reaching the vapor temperature corresponding to the boiling point of DMAc, the distillation was stopped, and the synthesis was carried out for 4 h. After that, the polymer solution was cooled to 90 °C, and 7 g of oxalic acid in 50 mL of DMAc was added. Then the polymer solution was filtered from the salts formed during the synthesis and precipitated by spraying into distilled water. The polymer was washed many times with hot distilled water and dried at 150 °C in a vacuum oven [25].

Modern methods of synthesis, which make it possible to obtain high-molecular compounds with desired properties and characteristics, are largely based on the rule of non-equivalence of functional groups. Stopping the growth of the polymer chain in nonequilibrium polycondensation when it is carried out in the presence of an excess of one of the monomers (Figure 1a) is caused by the fact that at a certain stage of the reaction, the resulting macromolecules will have the same functional groups of the excess component at both ends of the chain, excluding further elementary reaction acts leading to the growth of the polymer chain [26,27,28]:

In this work, the *M*_W_ of synthesized polymers was controlled by varying the molar ratio of 4,4′-dihydroxybiphenyl and 4,4′-dichlorodiphenylsulfone (DCDPS) monomers within 1:1–1.15 under the same temperature-time synthesis regime.

### 2.3. Study of Synthesized PPSUs

#### 2.3.1. Nuclear Magnetic Resonance (NMR) Method

High-resolution ^1^H NMR spectra were obtained for solutions in CDCl_3_ according to the standard procedure on a Bruker AVANCE III HD 400 NMR spectrometer.

#### 2.3.2. Gel Permeation Chromatography (GPC) Method

Gel-permeation chromatography (GPC) analysis of the polymers was also performed on a Waters system with a differential refractometer (Chromatopack Microgel-5; eluent, chloroform; flow rate, 1 mL/min). The molecular weights and polydispersity were calculated by a standard procedure relative to monodispersed polystyrene standards.

### 2.4. Preparation of Casting Solutions

Fifteen casting solutions of various compositions with a volume of 8 mL were prepared (Table 2). The components of the solution were mixed in glass vials at 20 °C for 24 h.

### 2.5. Study of Viscosity of Casting Solutions

To measure the viscosity of polymer solutions, a Brookfield DV III-Ultra rotational viscometer was used. Measurements for each solution were carried out at a temperature of 22 °C.

### 2.6. Sample Casting of Hollow Fiber Membranes

An original device based on a 3D printer, shown in Figure 2 [29], was used for casting samples of hollow fiber membranes.

The device includes a block (1) with a carrier needle (2) moving along vertical (6) and horizontal (7) bars in x, z coordinates. Samples of HF membranes were formed by successively lowering the carrier needle into the bottles (4) with the spinning solution and then into the bottles (4) with the precipitant. The bottles with casting solutions and precipitants are located on the polymer platform (3), which also moves along the bar (5) in the y coordinates.

The created software allows you to set the speed of movement of the carrier needle along the x, y, z coordinates and platforms with containers for various purposes (casting solutions, precipitants, etc.), as well as the contact time of the carrier needle with the casting solution, air and precipitant. The “wet” method of casting HF membranes was carried out using a carrier needle.

In this work, the speed of movement of the carrier needle along the x, y, z axes was selected relative to the viscosity of the polymer solution. The maximum velocity along the z axis was 20 mm/s, and along the x and y axes, 50 mm/s. Distilled water was used as a precipitant. Scanning electron microscopy (SEM) was used to characterize the structure and morphology of the membranes. SEM was carried out on a Thermo Fisher Phenom XL G2 Desktop SEM (Waltham, MA, USA). Cross-sections of the membranes were obtained in liquid nitrogen after preliminary impregnation of the specimens in isopropanol. A thin (5–10 nm) gold layer was deposited on the prepared samples in a vacuum chamber (~0.01 mbar) using a desktop magnetron sputter “Cressington 108 auto Sputter Coater” (Rassendale, Liverpool, UK). The accelerating voltage during image acquisition was 15 keV. Further image analysis and determination of the selective layer thickness were carried out using the Gwyddion software (version 2.53).

### 2.7. Ultrafiltration Properties

The ultrafiltration experiment was carried out on the setup described in Ref. [25]. The sample was sealed into the module. The measurement was performed in the flow mode with a transmembrane pressure of 1 bar.

The permeance was calculated using the formula:(1)$$P=\frac{V}{S\;\cdot t\;\cdot \;\Delta p}$$
where *P* is the permeance (L/m^2^·h bar), *V* is the volume of the taken sample (L), *t* is the sampling time (h), *S* is the surface area of the selective layer of the short-sample of the hollow fiber (m^2^), Δ*p* is the overpressure (bar).

The calculation of the rejection was carried out according to the formula:(2)$$R=\frac{1-C_{p}}{C_{f}}\cdot 100\;\%$$
where *R* is the rejection (%), *C_p_* is the concentration of the solute in the permeate (mg/L), and *C_f_* is the concentration of the solute in the feed stream (mg/L).

To measure the rejection coefficient, an aqueous solution of Blue Dextran (MM = 69 kg/mol) with a concentration of 100 mg/kg was prepared and used.

## 3. Results and Discussion

### 3.1. NMR of Synthesized PPSUs

The chemical structure of the synthesized polymers was studied by the NMR method. Figure (Figure 1b) shows the structure of the terminal fragment of the PPSU chain, determined by NMR studies.

The proton signals of the studied polymers are in the region of 6.8–8.2 ppm. The general view of the ^1^H spectrum is shown in Figure 3. The assignment of signals to the corresponding groups is given in Table 3.

The chain length was calculated by comparing the normalized (per one proton) signal intensities of the inner and terminal groups. The estimated number of links in the polymer chain is shown in Table 4.

The study of the molecular weight characteristics of the synthesized polymers by gel permeation chromatography showed that the synthesis of PPSU with blocking of end groups led to the production of polymers with a unimodal molar mass distribution. At the same time, as can be seen from Table 4 and Table 5, with an increase in the excess of DCDPS, the molecular weight of the polymers monotonously decreases. The results of gel permeation chromatography (GPC) are presented in Table 5. The decrease in the molecular weight of the polymers with increasing excess of DCDPS is confirmed both by the GPC method and by estimates made based on the NMR results. However, there is only a qualitative agreement between these two methods. This may be due to the high error in estimating the relative number of end groups by NMR. This is especially true for polymers with high molecular weight when the signal intensity of the terminal units is low. 

### 3.2. Study of the Rheological Properties of PPSU Solutions

The viscosity of spinning solutions is a key parameter in the spinning of hollow fiber membranes. Therefore, solutions with a concentration of 20 wt. % in the NMP standard for forming membranes and their viscosities were investigated. Figure 4 shows that with an increase in the molecular weight of the polymer, a viscosity of 20 wt. % solution increases from 340 to 12,100 mPa∙s. The viscosities used to produce HF membranes lie in the range of 10,000–60,000 mPa∙s. It can be seen that the viscosity reaches more than 10,000 mPa·s only for the PPSU 5 solution. Lower molecular weight polymers are not suitable for forming hollow fiber membranes.

To reach the required viscosity of the solution, the rheological properties of PPSU-5 solutions were studied in the concentration range of 15–24 wt. %. The graph shown in Figure 5 shows that with an increase in the polymer concentration, the viscosity of PPSU solutions increases by more than 10 times.

It is known that to increase the porosity and hydrophilicity of porous PPSU membranes, polyethylene glycol (PEG) is added to the casting solution [15]. We added PEG 400 to the casting solution and varied its concentration from 20 to 30 wt. %. The dynamic viscosity of these solutions is shown in Figure 6. As can be seen from Figure 6, the addition of PEG leads to an increase in the viscosity of the spinning solution. This result gives us the opportunity to work in the region of lower concentrations of PPSU 18–20 wt. %, which in the future will make it possible to form membranes with high porosity. Thus, the introduction of 20 wt. % PEG400 increased the viscosity to 14,400 and 17,500 mPa·s for 18 and 20 wt. % PPSU solution. Solutions of PPSU 18 and 20 wt. with 25 wt. % PEG400 had a viscosity of 28,500 and 47,600 mPa·s, with 30 wt. % PEG400—31,600 and 73,800 mPa·s, respectively.

### 3.3. Porous Structure, Transport and Separation Characteristics of Samples of PPSU-Based Hollow Fiber Membranes Spinning on the Needle-Carrier of the Device

Based on the prepared solutions of PPSU-5, using a manipulator, samples of hollow fiber membranes were formed by wet spinning. Figure 7 shows the SEM of a cross-section of hollow fiber membranes. Membrane formed from 20 wt. % solution of PPSU without the addition of PEG400 has a dense sponge structure with a selective layer thickness of about 20–25 µm (Figure 7a). For this reason, it has poor ultrafiltration properties (Table 6). Despite the high rejection, the fiber formed without the use of a pore-forming agent has a very low water permeance of about 0.2 L/m^2^·h bar, which does not correspond to the level of ultrafiltration membranes. 

With the introduction of PEG400 into the spinning solution, the permeance of membranes sharply increases by more than two orders of magnitude. At the same time, the membranes have high rejection.

Photographs of scanning electron microscopy (SEM) of hollow fiber membranes obtained on the basis of PPSU solutions containing PEG are shown in Figure 7b,c. When considering photographs of membranes prepared from a solution of 18% wt PPSU, as the concentration of PEG400 increased, a transition from a finger-like structure to a spongy one was observed. It should be noted that in the 25 wt. % PEG400 sample, finger-like pores become thinner, and their number increases relative to the sample from 20 wt. % PEG400. While in the sample with 30 wt. % PEG400, the formation of macrovoids in a dense spongy structure is observed. The formation of such a dense structure in the case of 30 wt. % PEG400 led to a sharp decrease in water permeance to 6.3 L/m^2^·h bar (Table 6) wherein a sample of 18 wt. % PPSU with 30 wt. % PEG400 showed a high rejection of Blue Dextran (*M*_W_ = 69,000 g/mol) 99.9%, which corresponds to trade-off permeance-selectivity.

For membranes formed from solution with PPSU concentration of 20 wt. %, narrowing of finger-shaped pores was observed, as well as an increase in their number with an increase in the PEG400 concentration in the spinning solution. More open finger-shaped pore structure leads to an increase in the water permeance of membranes from 36.6 to 95.7 L/m^2^·h bar with an increase in the concentration of PEG 400 from 20 to 30 wt. %. At the same time, all membranes have a high Blue Dextran retention rate of 99.9%.

The resulting ultrafiltration membranes from high molecular weight PPSU have competitive filtration properties relative to membranes formed from commercial PPSUs with a molecular weight of 50,000–60,000 g/mol. Therefore, the membrane developed in this work from solution containing 20 wt. % PPSU-5 with the addition of 20 wt. % PEG400 shows a water permeance almost two times higher than a membrane formed from 20 wt. % solution of pure PPSU (Solvay, Alpharetta, GA, USA) [11]. At the same time, the membrane from this work also demonstrates the best rejection of 99.9% compared to 93.9% in [11] for a model substance with *M*_W_ = 69,000 g/mol. Our membrane demonstrates water permeance three times higher than the membrane in the work [20] from 25 wt. % solution of sulfonated PPSU (Solvay Specialty Polymers (Bollate (MI)—Italy) with the addition of titanium dioxide nanoparticles (permeance of 28 L/m^2^·h bar). It has a close permeance value to the membrane obtained in the work [19] from 16 wt. % PPSU (Solvay Advanced Polymer, Brussels, Belgium) with the addition of tin dioxide nanoparticles (Table 1).

Thus, the introduction of PEG400 into the spinning solution makes it possible to radically change the filtration properties of membranes and create highly permeance ultrafiltration membranes based on PPSU.

## 4. Conclusions

In the work, PPSUs with different molecular weights were synthesized for the development of highly permeable ultrafiltration hollow fiber membranes for the first time. The *M*_W_ of synthesized polymers was controlled by varying the molar ratio of 4,4′-dihydroxybiphenyl and 4,4′-dichlorodiphenylsulfone monomers within 1:1–1.15 under the same synthesis conditions. The chemical structure of synthesized polymers was studied by NMR spectroscopy. The decrease in the molecular weight of the polymers with an increase in the excess of DCDPS monomer was proven both by the GPC method and by calculations based on NMR data. Based on the study of the rheological properties of polymer solutions, a high molecular weight PPSU (*M*_W_ = 102,000 g/mol) was chosen for the formation of hollow fiber membranes. In order to form membranes with higher porosity and a hydrophilized surface, the pore-forming agent PEG400 was added to the spinning solution. The addition of PEG to the spinning solution led to an increase in viscosity, which makes it possible to form membranes in the region of lower PPSU concentrations of 18–20 wt. %. Such membranes had a looser structure with higher porosity. Hollow fiber membranes were formed from PPSU solutions by the wet method using an express manipulator. With the addition of PEG400 to the spinning solution, the membrane permeance increased sharply by more than two orders of magnitude (from 0.2 to 96 L/m^2^·h bar). At the same time, the membranes had high rejection coefficients (99.9%) of Blue Dextran model filtered substance (*M*_W_ = 69,000 g/mol).

## Figures and Tables

**Figure 1 membranes-12-01113-f001:**
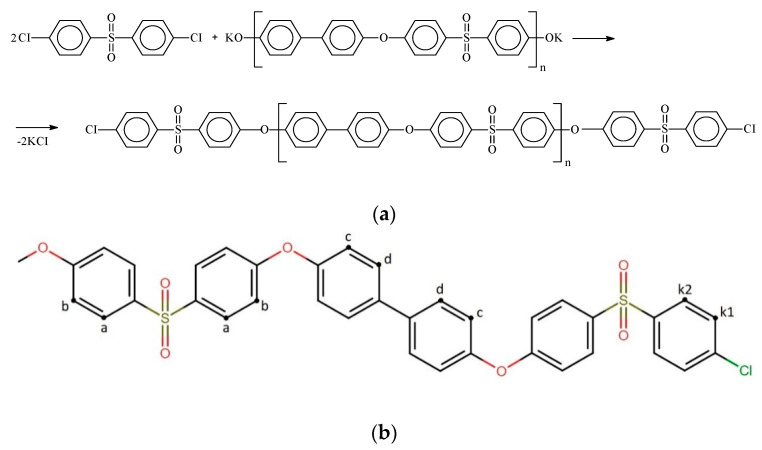
Scheme for the synthesis of PPSU (**a**), structure of the terminal fragment of the PPSU polymer chain (**b**).

**Figure 2 membranes-12-01113-f002:**
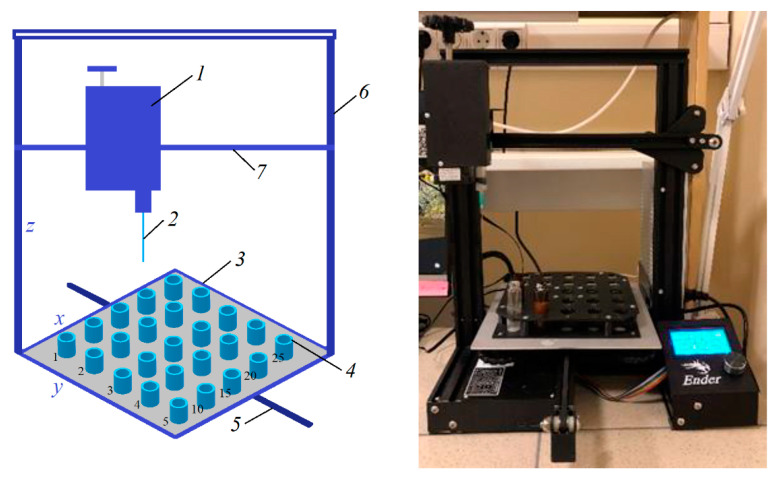
The original 3D printer used to cast PPSU hollow fiber membrane samples.

**Figure 3 membranes-12-01113-f003:**
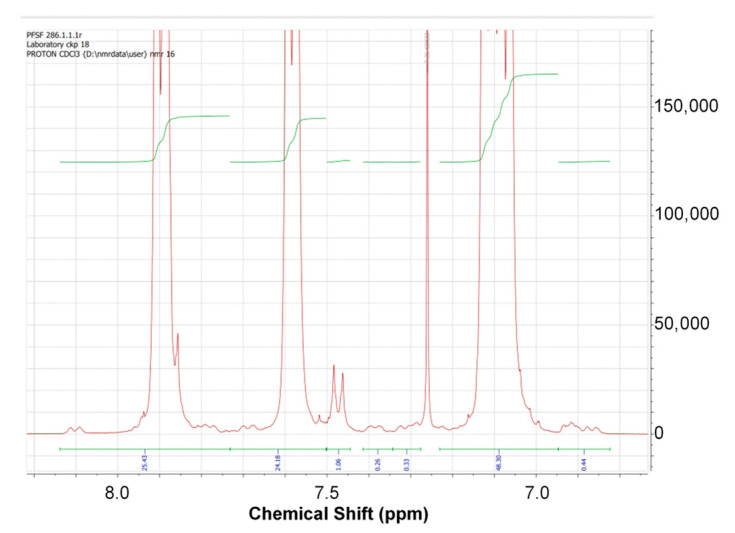
General view of the proton NMR spectrum.

**Figure 4 membranes-12-01113-f004:**
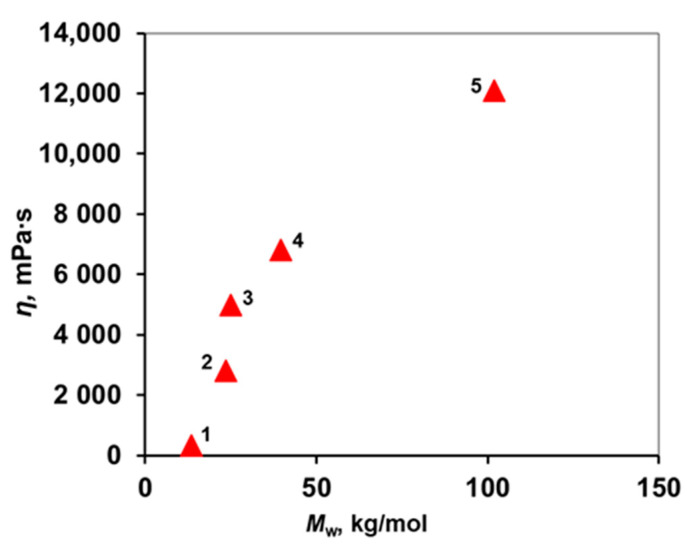
Dependence of the viscosity of PPSU 20 wt. % solutions in NMP on their molecular weight at 22 °C.

**Figure 5 membranes-12-01113-f005:**
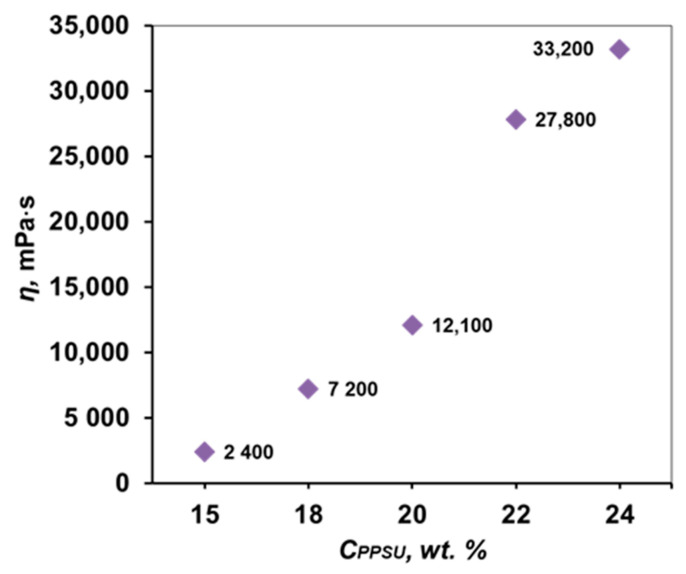
Dependences of the viscosity of PPSU 5 solutions in NMP on its concentration at 22 °C.

**Figure 6 membranes-12-01113-f006:**
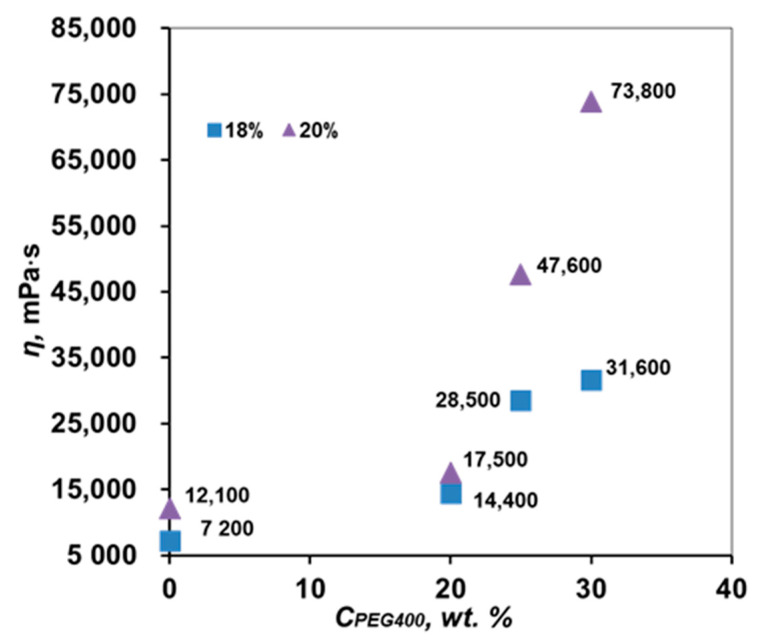
Dependence of the dynamic viscosity of the PPSU-5/NMP/PEG400 system on the PEG400 concentration at 22 °C.

**Figure 7 membranes-12-01113-f007:**
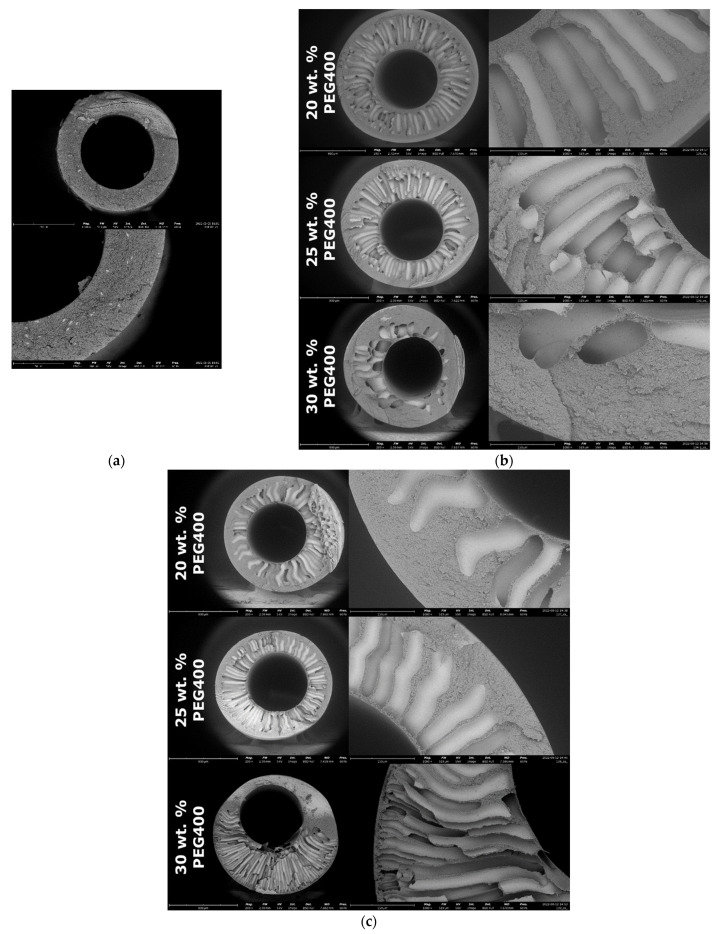
SEM photos of hollow fiber membranes obtained from PPSU-5 casting solutions: (**a**) 20 wt. % PPSU-5 hollow fiber without PEG; (**b**) 18 wt. % PPSU-5 hollow fiber with PEG400 added; (**c**) 20 wt. % PPSU-5 hollow fiber with PEG400 added.

**Table 1 membranes-12-01113-t001:** Literature data on UF membranes based on PPSUs.

Type of Polymer	Manufacture	*M_W_*_P_, g/mol	Type of Membranes	*C_P_*, mas. %	Additive	*P*, L/m^2^·h bar	Retained Substance	*M_W_*_S_, g/mol	*R*, %	Ref.
PPSU	Solvay, Alpharetta, GA, USA	50,000	HF	20	-	55.8	BSA	69,000	93.9	[11]
PPSU	BASF, Ludwigshafen, Germany	51,000	flat	16	-	36–488.4	-	-	-	[14]
PPSU	BASF, Ludwigshafen, Germany	48,000	flat	20	PEG6000	486	HSA	66,400	90	[15]
sPPSU	synthesis within the work	the source does not indicate	flat	17	PVP	250	BSA	67,000	92.2	[16]
PPSU	Solvay, Brussels, Belgium	50,000	flat	20	PVP	54	Dextran	2100	>90	[17]
Al-MOF/ PPSU	Solvay, Alpharetta, GA, USA	65,000	flat	20	Al-MOF	47.9	Methyl violet	394	93.8	[18]
PPSU	Solvay, Brussels, Belgium	50,000	HF	16	nano-SnO_2_	121	PEG	20,000	90	[19]
sPPSU	Solvay (Bollate (MI)—Italy)	49,000–55,000	HF	25	nano-TiO_2_	28	Trypsin	20,000	>90	[20]
PPSU	BASF, Ludwigshafen, Germany	51,000	HF	-	multiblock copolymers based on Lutensol^®^ AT80, Pluronic^®^ F127 or Pluriol^®^ E8000; PVP	~520	-	-	-	[21]
PPSU	BASF, Ludwigshafen, Germany	51,000	HF	17.4	multiblock copolymers based on Lutensol^®^ AT80, Pluronic^®^ F127 and Pluriol^®^ E8000; PVP; 1,2-propanediol	626	PEO	71,400	>90	[22]
PPSU	Solvay, Brussels, Belgium	50,000	HF	20	chitosan-based nanoparticles, silver-loaded chitosan nanoparticles	56.9	Reactive Black 5	991.8	89.3	[23]

**Table 2 membranes-12-01113-t002:** Compositions of spinning solutions.

PPSU	*C_P_*, wt. %	Pore-Forming Agent	*C_A_*, wt. %
1	20	-	-
2			
3			
4			
5			
5	15		
	18		
	22		
	24		
	18	PEG400	20
			25
			30
	20		20
			25
			30

**Table 3 membranes-12-01113-t003:** Assignment of proton signals to the corresponding groups.

Group	Chemical Shift (ppm)	Multiplicity
a	7.89	d
b	7.07	d
c	7.12	d
d	7.58	d
k1	7.47	d
k2	7.87	d

**Table 4 membranes-12-01113-t004:** Approximate number of units determined by NMR spectroscopy.

Sample Number	Estimated Number of Links
1	9
2	19
3	30
4	33
5	38

**Table 5 membranes-12-01113-t005:** Molecular weight and molar mass distribution of PPSUs determined by the GPC method and by estimates made based on the NMR results.

Sample Name PPSUs	*M*p × 10^−3^	*M*w × 10^−3^	*M*_N_ × 10^−3^	*M*w/*M*_N_	*M*_NMR_ × 10^−3^
1	9.4	13.4	4.0	3.3	3.6
2	21.2	23.5	11.6	2.0	7.6
3	26.7	25.0	5.2	4.8	12.0
4	34.5	39.5	20.7	1.9	13.2
5	84	102	38	2.7	15.2

**Table 6 membranes-12-01113-t006:** The results of measurements of the permeance and rejection of the model dye Blue Dextran.

*C*_PPSU_, wt. %	*C*_PEG400_, % Macc.	*P*, L/m^2^·h bar	*R*_Blue Dextran_ 69,000, %
18	20	15.7	78.6
18	25	103.6	87.8
18	30	6.3	99.9
20	0	0.2	99.9
20	20	36.6	99.9
20	25	59.9	99.9
20	30	95.7	99.9

## Data Availability

Not applicable.

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
