# Peer review of "Effect of Composition and Viscosity of Spinning Solution on Ultrafiltration Properties of Polyphenylene Sulfone Hollow-Fiber Membranes"

_membranes, 2022, doi:10.3390/membranes12111113_

Round 1

Reviewer 1 Report

1. The punctuation in lines 75-76 and the unit format in lines 81-82 are wrong.

2. For the "latest work" described in line 84, please add the latest progress in this area in 2022. 3. In the "Manufacture" column of Table 1, all are expressed in English. Please complete the data in the column "МWP, g/mol". Please unify the description in the "Additive" column. Delete the parameter column that does not support this article. References could be added.

4. The subheadings on lines 170 and 181 need to be re-condensed.

5. The number "platform3" in Figure 1 indicates repetition. Please add the actual picture of the device.

6. "table 2" repeats. Also, please optimize the first "table 2" form.

7. It is recommended to delete the SEM function description in subsection 2.7.1.

8. It is recommended to adjust the content of 3.1 to subsection 2.2.

9. Please integrate Figure 2 and Figure 3 into one picture for explanation.

10. Inappropriate reference to [15] in line 262.

11. Please add SEM images of different types of fiber membranes in Figure 8. In addition, the performance characterization process diagram of the film is added.

12. Please focus on the root cause of the different water permeability and interception rate of three different fibers according to Figure 8.

Author Response

Review Report

  1. The punctuation in lines 75-76 and the unit format in lines 81-82 are wrong.

Thanks for the comments. Corrected.

  1. For the "latest work" described in line 84, please add the latest progress in this area in 2022.

When searching for scientific publications in Scopus and Google Schoolar, no publications were found for 2022 on PPSU membranes for both filtration and other membrane processes. All publications are devoted to another polymer - polyphenylene sulfide.

  1. In the "Manufacture" column of Table 1, all are expressed in English. Please complete the data in the column "МWP, g/mol". Please unify the description in the "Additive" column. Delete the parameter column that does not support this article. References could be added.

Thanks for the comments. Changes made according to comments. Irrelevant columns have been removed.

  1. The subheadings on lines 170 and 181 need to be re-condensed.

Thanks for the comments. Corrected.

  1. The number "platform3" in Figure 1 indicates repetition. Please add the actual picture of the device.

The actual picture of the device has been added to Figure 1.

  1. "table 2" repeats. Also, please optimize the first "table 2" form.

The numbering has been corrected. Table 2 is optimized.

  1. It is recommended to delete the SEM function description in subsection 2.7.1.

Section 2.7.1. transferred to the production of membranes.

Numbering corrected.

  1. It is recommended to adjust the content of 3.1 to subsection 2.2.

Synthesis description moved to section 2.2

  1. Please integrate Figure 2 and Figure 3 into one picture for explanation.

Figures 2 and 3 are integrated.

  1. Inappropriate reference to [15] in line 262.

The commentary on the reference [15] has been corrected.

  1. Please add SEM images of different types of fiber membranes in Figure 8. In addition, the performance characterization process diagram of the film is added.

SEM image of a hollow fiber prepared from a PEG-free solution added to Figure 8. Figure 6 added.

  1. Please focus on the root cause of the different water permeability and interception rate of three different fibers according to Figure 8.

Section 3 has been substantially revised in accordance with the comments of the reviewer.

Reviewer 2 Report

In this manuscript, Anokhina et al. reports a high-performance ultrafiltration hollow fiber PPSU membrane, by regulating the molecular weight of synthesized PPSU and adding PEG400 into spinning solution to control the viscosity. The work is of interest. However, many descriptions in Results and Discussion do not match the experimental data, and there are many shortcomings in the present form. Thus, this manuscript needs to be improved before publication in this journal.

1) Page 2, line 84-89: “In the latest works of 2020, to obtain high-performance hollow fiber (HF) membranes…” and “At the same time, one work is described in the literature…”, please add reference.

2) Please check the sequence number of subheadings. For example, “2.2. Study of synthesized PPSUs”; “2.2.1. Nuclear magnetic resonance (NMR) method”; “2.2.2. Gel Permeation Chromatography (GPC) Method”; and “2.3. Preparation of spinning solutions”.

Moreover, please check the sequence number of Tables.

3) Where is the description for Figure 2 in manuscript?

4) In Figure 5, only PPSU 1 sample’s viscosity reaches more than 15000 mPa∙s and the viscosity of sample PPSU 5 is almost zero, these do not match the authors’ descriptions.

In Figure 6, the authors choose PPSU 5 to study rheological properties in the concentration range of 15-24 wt%, but when the concentration is 20 wt%, the viscosity is 12100, this does not match in Figure 5.

Moreover, for all experiments and discussions about PPSU with the addition of PEG400, the authors choose PPSU-5 as the solution, but in Table 1 the PPSU-1 is the solution.

5) Page 7, line 283-286: the authors describe “Although 20 wt. % solution of PPSU without the addition of PEG400 has sufficient viscosity to form a hollow fiber, membranes based on it do not have the necessary ultrafiltration properties.” When the PPSU solution is 20 wt%, the viscosity is 12100 as shown in Figure 5. However, in Page 5 line 246-249: the authors describe “a viscosity of 20 wt. % solution increases from 340 to 17,000 mPa∙s. This viscosity is not enough to form a mechanically strong hollow fiber. The viscosities used to produce HF membranes lie in the range of 15,000-60,000 mPa∙s.” These descriptions are inconsistent.

6) As the authors described for Figure 8c, “narrowing of finger-shaped pores was observed with an increase in their number for samples with a matrix polymer concentration of 20 wt %. with an increase in the PEG400 concentration in the spinning solution. The formation of a developed structure of finger-like pores led to an increase in the water permeance of membranes from 36.6 to 95.7 l/m2 h bar”. However, in Table 5, the permeance from PPSU with 30% PEG400 is higher than that from PPSU with 25%. What is the reason?

7) There are some minor mistakes, For example:

Line 138: 16 casting solutions in Table 2?

Page 3, line 181: please remove “of”.

Author Response

In this manuscript, Anokhina et al. reports a high-performance ultrafiltration hollow fiber PPSU membrane, by regulating the molecular weight of synthesized PPSU and adding PEG400 into spinning solution to control the viscosity. The work is of interest. However, many descriptions in Results and Discussion do not match the experimental data, and there are many shortcomings in the present form. Thus, this manuscript needs to be improved before publication in this journal.

1) Page 2, line 84-89: “In the latest works of 2020, to obtain high-performance hollow fiber (HF) membranes…” and “At the same time, one work is described in the literature…”, please add reference.

References are added to the text of the article

2) Please check the sequence number of subheadings. For example, “2.2. Study of synthesized PPSUs”; “2.2.1. Nuclear magnetic resonance (NMR) method”; “2.2.2. Gel Permeation Chromatography (GPC) Method”; and “2.3. Preparation of spinning solutions”. Moreover, please check the sequence number of Tables.

The numbering of subheadings and tables is given in the correct order in accordance with the comments.

3) Where is the description for Figure 2 in manuscript?

The reference to Figure 2 has been added to the text of the article.

4) In Figure 5, only PPSU 1 sample’s viscosity reaches more than 15000 mPa∙s and the viscosity of sample PPSU 5 is almost zero, these do not match the authors’ descriptions.

The authors thank the reviewer. The figure shows incorrect numbering of polymers. The figure has been corrected.

In Figure 6, the authors choose PPSU 5 to study rheological properties in the concentration range of 15-24 wt%, but when the concentration is 20 wt%, the viscosity is 12100, this does not match in Figure 5.

The authors thank the reviewer. The figure 5 shows incorrect viscosity for PPSU 5. The figure has been corrected.

Moreover, for all experiments and discussions about PPSU with the addition of PEG400, the authors choose PPSU-5 as the solution, but in Table 1 the PPSU-1 is the solution.

The authors thank the reviewer. The table shows incorrect number of polymer. The table has been corrected.

5) Page 7, line 283-286: the authors describe “Although 20 wt. % solution of PPSU without the addition of PEG400 has sufficient viscosity to form a hollow fiber, membranes based on it do not have the necessary ultrafiltration properties.” When the PPSU solution is 20 wt%, the viscosity is 12100 as shown in Figure 5. However, in Page 5 line 246-249: the authors describe “a viscosity of 20 wt. % solution increases from 340 to 17,000 mPa∙s. This viscosity is not enough to form a mechanically strong hollow fiber. The viscosities used to produce HF membranes lie in the range of 15,000-60,000 mPa∙s.” These descriptions are inconsistent.

The authors thank the reviewer. On the Page 7 corrections have been made.

6) As the authors described for Figure 8c, “narrowing of finger-shaped pores was observed with an increase in their number for samples with a matrix polymer concentration of 20 wt %. with an increase in the PEG400 concentration in the spinning solution. The formation of a developed structure of finger-like pores led to an increase in the water permeance of membranes from 36.6 to 95.7 l/m2 h bar”. However, in Table 5, the permeance from PPSU with 30% PEG400 is higher than that from PPSU with 25%. What is the reason?

The table contained a typo. Corrections have been made to the text of the article.

7) There are some minor mistakes, For example:

Line 138: 16 casting solutions in Table 2?

Page 3, line 181: please remove “of”.

The authors thank the reviewer. Corrections have been made to the text of the article.

The authors are grateful to the reviewer for the comments! The article has become much improved and more attractive to the reader.

Round 2

Reviewer 1 Report

 1、The number "platform3" in Figure2  indicates repetition. Please add the actual picture of the device. 

2、Please refine the conclusion and summary again!

Author Response

1、The number "platform3" in Figure 2  indicates repetition. Please add the actual picture of the device.

Note corrected. Added to figure two is a real image of the device.

2、Please refine the conclusion and summary again!

The abstract and conclusions have been clarified.

Reviewer 2 Report

The manuscript is significantly improved. I think this manuscript could be published in this journal.

Author Response

Thanks to the reviewer for the positive response.